# OpenReview forum: "Thought Branches: Interpreting LLM Reasoning Requires Resampling"
_ICLR.cc/2026/Conference — ICLR 2026 Poster_

### Official Review · Reviewer_CEUK · 2025-10-31

**Soundness:** 3
**Presentation:** 3
**Contribution:** 3
**Rating:** 6
**Confidence:** 3

**Summary:**

This paper argues that interpreting reasoning LLMs requires studying distributions of chain-of-thought (CoT) trajectories rather than single rollouts. The authors introduce on-policy resampling methods with two key innovations: (1) a resilience metric measuring how many resamples are needed to eliminate semantic content (addressing error correction), and (2) counterfactual++ importance measuring causal impact when content is fully absent. Applied to four reasoning models across three domains, main findings include: normative self-preservation statements show low resilience and minimal causal impact in blackmail scenarios; on-policy resampled interventions produce substantially larger effects than off-policy edits (up to 100% vs. near-zero changes); and unfaithful reasoning exhibits smooth, cumulative biases ("nudged reasoning") rather than discrete pivots, demonstrated in both hinted MMLU questions and biased resume

**Strengths:**

The conceptual shift from single-trace to distributional analysis is timely and well-motivated for stochastic reasoning systems. The resilience metric elegantly addresses a real problem (error correction/semantic regeneration) that prior ablation methods ignore. The on-policy vs. off-policy distinction provides theoretical grounding for why previous CoT intervention work showed small effects.

The methodology is principled: resilience measurement (Algorithm 1) and counterfactual++ importance properly handle semantic persistence across rollouts. Testing across multiple models (4 reasoning models) and domains (blackmail, MMLU, resume screening, MATH) demonstrates thoroughness. The on-policy intervention comparison (Section 3) is particularly well-instrumented with both global effects (Figure 3) and local plan changes (Table 1).

Addresses important safety questions around CoT monitoring and agentic misalignment. The finding that on-policy interventions outperform off-policy edits has direct implications for CoT steering research. The "nudged reasoning" characterization of unfaithfulness is more nuanced than prior binary (faithful/post-hoc) framings.

**Weaknesses:**

Limited generalizability. Core causal findings rest on one blackmail scenario with only 20 base traces. Authors acknowledge "largely hone into specific prompts rather than being prompt-agnostic". Unclear whether resilience patterns, self-preservation findings, or on-policy effectiveness generalize to: shorter CoTs, tool-augmented agents, production systems, other misalignment scenarios, or different prompt styles.

Validation gaps. Heavy reliance on GPT-4.1-mini for auto-labeling sentence categories and judging blackmail responses without human validation. Since effect sizes are small, modest classification errors could support same conclusions for wrong reasons.

Resilience interpretation confound: If models repeatedly regenerate content, does this indicate the idea is causally necessary OR that it's redundant (many paths to same conclusion)? These have opposite interpretations.

On-policy filtering is itself guided editing: The semantic similarity filter constrains the distribution being sampled. How much does this "care" in construction drive the superiority over off-policy? A soft steering baseline would strengthen claims.

**Questions:**

Sensitivity analysis: Please provide results varying the semantic similarity threshold (25th, 50th, 75th percentiles). How robust are Figures 1-2 and the self-preservation conclusions to this choice?

Validation of auto-labeling: What is the inter-rater reliability between your GPT-4.1-mini judge and human annotators for (a) sentence category labels and (b) blackmail classifications? Please provide validation on at least a subset of data.

Generalization evidence: Can you apply your methods to at least 2-3 other misalignment or decision-making scenarios beyond the Lynch et al. blackmail prompt?

---

### Official Review · Reviewer_xzeN · 2025-10-31

**Soundness:** 2
**Presentation:** 2
**Contribution:** 3
**Rating:** 4
**Confidence:** 3

**Summary:**

This paper argues that interpreting an LLM through a single chain of thought is insufficient, since models reason over distributions of possible trajectories. The authors propose two metrics, resilience, capturing how robustly a sentence’s meaning persists under on-policy resampling, and counterfactual++ importance, measuring its true causal influence on the model’s output. Applied to blackmail, math, and resume-screening tasks, the framework reveals that “self-preservation” sentences have little effect on decisions, on-policy interventions are more effective than off-policy edits, and subtle hints can gradually steer reasoning.

**Strengths:**

1) Strong intervention results as off-policy insertions cluster near zero effect, whereas on-policy resampled replacements produce substantially larger, directional changes with up to 100% reductions in blackmail rate.

2) The paper makes “reasoning-as-a-distribution” concrete by defining on-policy resampling and introducing two metrics, resilience and counterfactual++ to quantify sentence-level causal impact.

3) Provides a clear causal framework linking reasoning bias and unfaithful CoT, showing that small prompt nudges cause smooth, measurable shifts in model behavior.

**Weaknesses:**

1) The claim that reliable interpretability benefits from on-policy resampling is supported mainly by experiments rather than theory. It assumes that CoT tokens reflect mechanistic causality rather than surface-level narration, but the paper does not clarify when this assumption holds. While resampling outperforms off-policy edits in the blackmail task, results in the math domain show handwritten edits can be locally competitive, raising doubts about generality. The paper could formalize the causal estimand of on-policy resampling, e.g., specify a simple SCM over CoT tokens and decisions, and state assumptions under which counterfactual++ is consistent.

2) The evaluation scope remains narrow. The paper could include matched off-policy baselines and sensitivity analyses (e.g., temperature, random seeds) to test robustness.

3) The approach relies on SBERT embeddings to measure semantic similarity and cluster sentences, but these external encoders may not align with model-internal representations. This makes importance estimates sensitive to the embedding choice and thresholding. The paper could test robustness by varying the encoder or similarity cutoff and reporting whether conclusions remain consistent.

**Questions:**

1) The blackmail taxonomy is judged by an external LLM. Did you assess the quality? If so how?
2) Self-preservation sentences are defined via auto-labels. Could these sentences be downstream rationalizations but the upstream latent still encodes a survival goal?
3) The “nudged reasoning” story is shown on professor-hinted MMLU. Does it hold for subtle hints and for correct hints?
4) Sentence clusters (k=32) are formed per-resume. How stable are clusters under different k or different embedding models? Provide ARI across runs and qualitative analysis of cluster quality.
5) Typos:
Page 4: "the its upstream context"
Page 4: "is just examining"

---

### Official Review · Reviewer_Hxtb · 2025-11-01

**Soundness:** 3
**Presentation:** 3
**Contribution:** 3
**Rating:** 6
**Confidence:** 4

**Summary:**

This paper argues that interpreting a single CoT trace from a reasoning model is insufficient because CoTs represent just one sample from a distribution over possible reasoning trajectories. To address this, the authors propose on-policy resampling---regenerating continuations from intermediate steps---to approximate this distribution. They introduce metrics such as resilience and counterfactual++ importance to estimate sentence-level causal influence. The method is evaluated across three domains: a blackmail scenario, hinted multiple-choice questions, and a resume-screening task.

**Strengths:**

1. Novel framing of reasoning as distributions over trajectories instead of isolated CoTs.

2. novel metrics as well, such as resilience and counterfactual++ importance that address error-correction and trajectory divergence.

3. Rigorous experimental setup with 100+ rollouts per sentence and multiple reasoning models.

4. Important empirical result: self-preservation statements have low causal influence, challenging assumptions in AI safety.

5. Practical insight that off-policy edits underestimate causal effects compared to on-policy resampling.

**Weaknesses:**

1. Limited generalization beyond a few domains; resume experiments use only one model due to variability.

2. Heavy reliance on semantic similarity thresholds without justification or sensitivity analysis.

3. Causal claims are sometimes speculative; resampling may introduce distributional shifts rather than isolate causal effects.

4. No comparison with gradient-based or attention-based attribution methods.

5. Lacks evaluation of simpler baselines like random resampling or deletion-only methods.

6. Resilience algorithm stops only when semantic abandonment occurs; decay dynamics are not measured or analyzed.

7. Transplant resampling assumes nudged reasoning but could also reflect incremental post-hoc justification.

8. Resume experiment clustering uses 32 clusters without justification and lacks cross-resume comparability.

**Questions:**

1. Why use the median cosine similarity as a threshold for semantic equivalence? How sensitive are results to this choice?

2. Can you justify the assumption that on-policy resampling isolates causal effects rather than introducing new trajectory biases?

3. What criteria guided the selection of 32 clusters in the resume task? Would different cluster counts change conclusions?

4. What explains the remaining 22.5% demographic influence not mediated by explanation clusters?

5. In transplant experiments, how do you distinguish between nudged reasoning and incremental post-hoc rationalization?

6. Would your conclusions hold for domains with more formally checkable reasoning (e.g., theorem proving)?

---

### Official Review · Reviewer_BbjA · 2025-11-03

**Soundness:** 3
**Presentation:** 2
**Contribution:** 3
**Rating:** 6
**Confidence:** 3

**Summary:**

The paper proposes that interpreting LLM reasoning from a single chain-of-thought (CoT) is unreliable due to inherent stochasticity of CoTs, hence we should consider multiple rollouts. It introduces resampling-based metrics,  counterfactual importance and its variants counterfactual importance++, that explicitly incorporates resilience to changes as well. These metrics help to quantity the effect of sentence level changes in CoT on model's output distribution and attribute causal effects more reliably. They conduct analysis on a diverse set of tasks, agentic misalignment (blackmail) task, multiple-choice reasoning with hints, and resume evaluation, and present interesting findings that underscore the importance of taking several rollouts into consideration.

**Strengths:**

- The key framing of CoT interpretability over a distribution of rollouts and formalizing it through counterfactual importance and resilience metrics is well-motivated and novel to the best of my knowledge.

- The experimental scale and diversity lend to the technical soundness of the work. The analysis spans multiple large language models (Qwen, Llama, DeepSeek) and a variety of case studies. The experiments on analyzing the blackmailing task involved around 100 rollouts per base prompt, providing robust estimates for counterfactual importance metrics. Similarly, they considered several options for the off-policy interventions, conclusively showing that the efficacy of on-policy interventions.

- The resilience score is a valuable and significant contribution. It is important to ensure that a removed sentence does not reappear anywhere in the CoT trace. While the counterfactual score only checks for reoccurrence at the original edit location, its good to extend this check to future steps as well.

**Weaknesses:**

The key issue is with the clarity in writing and presentation of the work. Presentation needs to be improved as a lot details about the setup  are moved to the appendix. This makes it hard for someone who is not familiar with the exact prior tasks. For example, there is no description at all about the blackmailing task and its completely described in the appendix. A short summary about the task in the main text would be good. Further, the metric details were a little hard to follow in the main text. For section 4, it would have been nice if the authors can specify a small causal graph indicating what exactly is the mediator variable.

**Questions:**

- In the computation of counterfactual importance, how do you handle cases where, after removing a sentence $s_i$, the model regenerates the same or a semantically equivalent sentence during the counterfactual rollout? Are such rollouts included in the KL estimation, or are they filtered out via rejection sampling)? More generally, across the 100 rollouts per prompt, how many are ultimately retained for computing the metric, and do you discard the samples with similar sentence regenerated?

- What are the authors’ thoughts on intervening at different granularities within the CoT trace, rather than at the sentence level? For instance, could one perform rollouts around specific phrases or key words that represent important concepts, instead of entire sentences? More generally, would it make sense to first extract latent concepts from the CoT and intervene on those, to better capture their causal effect on the final outcome?

---

### Meta-Review · Area_Chair_nbVq · 2026-01-08

**Summary:**

This paper challenges the prevailing practice of interpreting Large Language Models (LLMs) based on single Chain-of-Thought (CoT) traces. The authors argue that because CoTs are stochastic, they should be viewed as distributions over trajectories. To operationalize this, the paper introduces a framework based on on-policy resampling. While the reviewers largely agreed that the conceptual shift from single-trace to distributional analysis is novel and timely, there were concerns regarding the robustness of the methodology and the scope of evaluation. Overall, since the authors fail to respond during the rebuttal phase, I am recommending an acceptance, but I wouldn't mind if the paper gets rejected.

**Reviewer Concerns:**

(1) Sensitivity and Robustness of Metrics (Major Concern) : Three reviewers (Hxtb, xzeN, CEUK) raised concerns about the technical implementation of the metrics. The method relies heavily on SBERT embeddings and specific cosine similarity thresholds (e.g., median) to determine if a thought has "reappeared." Reviewers noted a lack of sensitivity analysis. It is unclear if the findings hold if these thresholds are shifted (25th vs. 75th percentile). Reviewer xzeN noted that reliance on external encoders (SBERT) might not align with the model's internal representations, potentially introducing bias into the importance estimates.

(2) Generalizability and Evaluation Scope: The causal findings regarding "self-preservation" rely heavily on a single blackmail scenario with limited base traces. Reviewers (CEUK, Hxtb) questioned whether these patterns generalize to other misalignment scenarios, shorter CoTs, or tool-augmented agents. Reviewer Hxtb noted a lack of comparison with simpler baselines, such as random resampling or gradient/attention-based attribution methods, which makes it difficult to contextualize the performance of Counterfactual++.

**Reviewer Scores:**

All reviewers would keep their score since there is no rebuttal from the authors.

---

### Decision · Program_Chairs · 2026-01-26

Accept (Poster)